# Anti-Inflammatory Activity of 4-((1*R*,2*R*)-3-Hydroxy-1-(4-hydroxyphenyl)-1-methoxypropan-2-yl)-2-methoxyphenol Isolated from *Juglans mandshurica* Maxim. in LPS-Stimulated RAW 264.7 Macrophages and Zebrafish Larvae Model

**DOI:** 10.3390/ph14080771

**Published:** 2021-08-06

**Authors:** Su-Hyeon Cho, SeonJu Park, Hoibin Jeong, Song-Rae Kim, Myeong Seon Jeong, Miri Choi, Seung Hyun Kim, Kil-Nam Kim

**Affiliations:** 1Chuncheon Center, Korea Basic Science Institute (KBSI), Chuncheon 24341, Korea; chosh93@kbsi.re.kr (S.-H.C.); sjp19@kbsi.re.kr (S.P.); hbj04@kbsi.re.kr (H.J.); ksr87@kbsi.re.kr (S.-R.K.); jms0727@kbsi.re.kr (M.S.J.); mrchoi0829@kbsi.re.kr (M.C.); 2Department of Medical Biomaterials Engineering, College of Biomedical Sciences, Kangwon National University, Chuncheon 24341, Korea; 3College of Pharmacy, Yonsei Institute of Pharmaceutical Sciences, Yonsei University, Incheon 21983, Korea; 4Department of Bio-Analysis Science, University of Science & Technology, Daejeon 34113, Korea

**Keywords:** *Juglans mandshurica* Maxim., anti-inflammatory activity, NF-κB, MAPK, zebrafish larvae

## Abstract

*Juglans mandshurica* Maxim., a traditional folk medicinal plant, is widely distributed in Korea and China. In our previous study, we isolated a new phenylpropanoid compound, 4-((1*R*,2*R*)-3-hydroxy-1-(4-hydroxyphenyl)-1-methoxypropan-2-yl)-2-methoxyphenol (HHMP), from *J. mandshurica*. In the present study, we evaluated the anti-inflammatory activity of HHMP on lipopolysaccharide (LPS)-stimulated RAW 264.7 cells and zebrafish larvae. HHMP significantly inhibited LPS-induced nitric oxide (NO) and prostaglandin E_2_ production in a dose-dependent manner. Moreover, HHMP treatment considerably suppressed LPS-induced expression of inducible nitric oxide synthase and cyclooxygenase-2. We also demonstrated the mechanisms of HHMP inhibition of inflammatory responses in LPS-stimulated RAW 264.7 cells via Western blot analysis and immunofluorescence staining. Furthermore, HHMP significantly inhibited NO production in LPS-stimulated zebrafish larvae. Consequently, we established that HHMP significantly inhibited the LPS-induced activation of NF-κB and MAPK and the nuclear translocation of p65 in RAW 264.7 cells. Taken together, our findings demonstrate the effect of HHMP on LPS-induced inflammatory responses in vitro and in vivo, suggesting its potential to be used as a natural anti-inflammatory agent.

## 1. Introduction

Macrophages play a crucial role in the inflammatory response of the body by removing the initial cause of cell damage through the secretion of various inflammation-related factors, such as nitric oxide (NO), prostaglandin E_2_ (PGE_2_), pro-inflammatory cytokines, inducible nitric oxide synthase (iNOS), and cyclooxygenase (COX)-2 [1,2]. However, an excessive inflammatory response can be harmful and can affect the pathogenesis of many diseases, such as diabetes, arthritis, Alzheimer’s disease, cancer, and cardiovascular diseases [3,4]. External stimuli, such as lipopolysaccharide (LPS), induce the secretion of cytokines such as tumor necrosis factor (TNF)-α and interleukin-6, as well as enzymes such as iNOS and COX-2, leading to an inflammatory response [5]. Nuclear transcription factor kappa-B (NF-κB) and mitogen-activated protein kinase (MAPK) signaling pathways have been reported to regulate inflammatory reactions in LPS-stimulated macrophages [6,7]. NF-κB is a significant factor that manages immunity through the regulation of inflammation-related gene expression [8]. In LPS-stimulated RAW 264.7 cells, the phosphorylation of the inhibitor of NF-κB (IκB) by IκB kinase (IKK) activates NF-κB, which then translocates to the nucleus and promotes the expression of inflammation-related genes [7,9,10]. Moreover, MAPKs, which are composed of c-Jun N-terminal kinase (JNK), extracellular signal-regulated kinase (ERK or p42/p44), and p38, also cause the overexpression of inflammation-related mediators [11]. Therefore, it is important to explore compounds that suppress the activation of the NF-κB and MAPK signaling pathways.

In modern society, various steroidal and nonsteroidal drugs are used as anti-inflammatory medications; however, the various side effects caused by them have emerged as social problems [12,13]. Studies on anti-inflammatory compounds derived from natural products that can reduce these side effects have recently been actively conducted. The Manchurian walnut, *Juglans mandshurica* Maxim., is widely distributed and utilized as a traditional folk medicine to treat cancer, gastritis, dermatosis, and leucorrhea in Korea and China [14,15,16]. Previous studies have shown that *J. mandshurica* extract exhibits various biological effects, including the inhibition of allergic dermatitis-like skin disease, as well as anti-tumor and anti-diabetic activities [14,17,18]. Moreover, in our previous study, we isolated various compounds from *J. mandshurica* fruit and reported a new phenylpropanoid compound, 4-((1*R*,2*R*)-3-hydroxy-1-(4-hydroxyphenyl)-1-methoxypropan-2-yl)-2-methoxyphenol (HHMP), for the first time [19]. A previous study reported the inhibition of melanogenesis by HHMP in vitro [20]; however, no other activities of HHMP have been studied so far. Other phenylpropanoid compounds have been proven to inhibit inflammatory responses in vitro [21,22]; therefore, we investigated the protective effects of HHMP isolated from *J. mandshurica* on LPS-induced inflammatory response in RAW 264.7 cells and zebrafish larvae models.

## 2. Results

### 2.1. Effect of HHMP on LPS-Induced NO and PGE_2_ Production and Cell Viability in RAW 264.7 Cells

We determined whether HHMP suppresses LPS-induced NO and PGE_2_ generation in RAW 264.7 cells. LPS promoted NO and PGE_2_ production compared to the control group; however, HHMP significantly inhibited NO production (81.91%, 66.8%, and 48.98%) and PGE_2_ production (99.38%, 75.94%, and 54.86%) compared to the LPS-treated group in a dose-dependent manner (Figure 1D,E). Additionally, in the HHMP + LPS-treated group, HHMP did not affect cell viability compared to the LPS-treated group in RAW 264.7 cells (Figure 1B,C).

### 2.2. Effect of HHMP on LPS-Induced iNOS and COX-2 Protein Expression in RAW 264.7 Cells

We demonstrated whether the inhibitory activity of HHMP on NO and PGE_2_ production was exhibited through iNOS and COX-2 regulation. Although LPS stimulation induced iNOS expression levels, the addition of HHMP significantly reduced their expression levels in RAW 264.7 cells. Although the expression levels of COX-2 were unaffected at 12.5 and 25 μM HHMP, at 50 μM HHMP, the expression level of COX-2 was significantly inhibited (Figure 2).

### 2.3. Effect of HHMP on LPS-Induced NF-κB Activation in RAW 264.7 Cells

To further investigate the role of the NF-κB signaling pathway in the inhibition of the inflammatory response by HHMP, we examined the effect of HHMP on LPS-induced phosphorylation and translocation of NF-κB by Western blot analysis and immunofluorescence staining in RAW 264.7 cells. As shown in Figure 3A, significant differences were observed between the LPS-treated and HHMP + LPS-treated groups in the phosphorylation of NF-κB. Although treatment with LPS induced the phosphorylation of IκB, p65, and p105 compared to the control group, 50 μM HHMP markedly suppressed the LPS-induced phosphorylation of these proteins in RAW 264.7 cells. Moreover, LPS-induced nuclear translocation of p65 was repressed with HHMP treatment in RAW 264.7 cells (Figure 3B).

### 2.4. Effect of HHMP on LPS-Induced MAPK Activation in RAW 264.7 Cells

To prove the role of the MAPK signaling pathway in the inhibition of the inflammatory response via HHMP, the phosphorylation patterns of appropriate signaling molecules of the MAPK family were studied. The effect of HHMP on LPS-induced phosphorylation of ERK, JNK, and p38 was determined by Western blot analysis in RAW 264.7 cells. Consequently, as shown in Figure 4, significant differences were observed between the LPS-treated and HHMP + LPS-treated groups on the phosphorylation of ERK, JNK, and p38. When the cells were stimulated with LPS, the levels of phosphorylated ERK, JNK, and p38 increased compared to those in the control group. However, 50 μM HHMP significantly inhibited the phosphorylation of ERK, JNK, and p38 in RAW 264.7 cells.

### 2.5. Effect of HHmP on Survival Rate and NO Production in LPS-Stimulated Zebrafish Larvae

To examine the effect of HHMP on LPS-induced inflammation in zebrafish larvae, we measured the survival rate and NO production. Results showed that the survival rate was more than 80% for all tested concentrations of HHMP compared to the control group, indicating that they were non-toxic to the zebrafish larvae (Figure A). Furthermore, while treatment with LPS generated NO compared to the control group, 50 μM of HHMP significantly inhibited LPS-induced NO production in zebrafish larvae (Figure 5).

## 3. Discussion

Owing to the presence of various bioactive constituents and their stability, many researchers have explored biologically active natural products, including those that have anti-inflammatory properties [23,24,25]. In particular, phenylpropanoid compounds isolated from natural products are known to exhibit anti-inflammatory activities [21,26]. We isolated a new phenylpropanoid compound, 4-((1*R*,2*R*)-3-hydroxy-1-(4-hydroxyphenyl)-1-methoxypropan-2-yl)-2-methoxyphenol (HHMP), from the fruit extract of *J. mandshurica* in our previous study [19]. However, the inhibitory activity of HHMP against the inflammatory response has not yet been determined. Therefore, we identified the anti-inflammatory activity of HHMP against the LPS-induced inflammatory response in RAW 264.7 cells.

Macrophages cause inflammatory response when exposed to bacterial cell wall components, such as LPS [27]. LPS-stimulated macrophages secrete various inflammation-related factors, including NO, PGE_2_, cytokines, and enzymes [27,28]. These mediators promote the progression of inflammatory response and exacerbate inflammation through synergistic interactions with other inflammation-related mediators [27,29,30]. Consequently, these interactions induce acute and/or chronic inflammatory responses and diseases in tissues, including those of the heart, liver, lung, kidney, pancreas, brain, intestinal tract, and reproductive system [31]. Therefore, the investigation of anti-inflammatory bioactive compounds derived from natural products plays an important role in preventing and treating inflammatory diseases to maintain human health.

First, to prove the effect of HHMP on LPS-induced inflammatory response, we assured that HHMP concentrations of 12.5, 25, and 50 μM did not exhibit cytotoxicity against RAW 264.7 cells and used these concentrations for subsequent experiments. Subsequently, we confirmed that HHMP significantly inhibited NO generation by controlling the expression of the iNOS in LPS-stimulated RAW 264.7 cells. Moreover, we proved that PGE_2_ production was significantly inhibited in a concentration-dependent manner. However, 12.5 and 25 μM HHMP exhibited no significant difference in LPS-induced COX-2 expression levels compared with the LPS-treated group. Moreover, 50 μM HHMP significantly inhibited COX-2 expression levels. These results indicate that HHMP reduces NO and PGE_2_ production by inhibiting LPS-induced expression of the inflammatory enzymes iNOS and COX-2. These results also correspond with previous studies that have shown that low concentrations result in different tendencies of PGE_2_ production and COX-2 expression [32,33,34]. Therefore, our results showed that low concentration (12.5 and 25 μM) of HHMP inhibits PGE_2_ production without affecting COX-2.

Furthermore, we performed a mechanistic study on the anti-inflammatory activity of HHMP in LPS-stimulated RAW 264.7 cells. NF-κB controls the expression of various genes that play important roles in cell proliferation, tumorigenesis, and inflammation [35,36]. It plays a crucial role as a transcription factor in the inflammatory response and is sequestered in the cytosol as an inactive precursor protein, combined with IκB [36,37]. Stimulation with LPS causes the activation of IKK, and then rapid phosphorylation, ubiquitination, and proteolytic degradation of IκB [36,37,38,39]. Furthermore, the precursor protein NF-κB1 (p105) is activated, ubiquitinated, and degraded by proteasome, thereby yielding p50 [39]. This process leads to the activation of the NF-κB heterodimer (p50/p65) and translocation into the nucleus [33,37]. Consequently, nuclear translocation causes its binding to the target sites and promotes the transcription of pro-inflammatory mediators and cytokines [8,36]. HHMP inhibited the phosphorylation of IκB, p105, and p65, as well as the nuclear translocation of p65 in LPS-stimulated RAW 264.7 cells. Therefore, we demonstrated that the anti-inflammatory effect of HHMP occurs by inhibiting inflammation-related gene expression by blocking the NF-κB signaling pathway.

MAPKs containing p38, JNK, and ERK have crucial functions in physiological responses, including cell proliferation, differentiation, development, apoptosis, and inflammation in mammalian cells [36,40]. They are the major elements in the signaling pathway that cause the expression of pro-inflammatory mediators [41]. Moreover, they are involved in the posttranslational control of NF-κB and activator protein (AP)-1 [42]. The activation of MAPKs is induced by LPS stimulation and is associated with the overexpression of iNOS, COX-2, and pro-inflammatory cytokines [36,43]. Therefore, we assessed the MAPK signaling pathway to demonstrate the mechanism underlying the anti-inflammatory activity of HHMP in LPS-stimulated RAW 264.7 cells. The present results demonstrated that HHMP significantly suppressed the phosphorylation of ERK, JNK, and p38 in LPS-activated RAW 264.7 cells, indicating that HHMP significantly suppresses inflammation-related gene expression via the MAPK signaling pathway.

Due to the numerous advantages of the zebrafish model, including low husbandry costs, small size, optical transparency, and rapid development, it has been used in various research fields [44,45,46]. In previous studies, many researchers have used zebrafish as an animal model to report anti-inflammatory activity that could be analyzed in vivo rapidly and easily [47,48,49]. Therefore, we examined the inhibitory activity of HHMP on LPS-induced inflammation in zebrafish larvae. All concentrations of HHMP were non-toxic; hence, we used 50 μM HHMP to measure NO inhibitory activity. Treatment with 50 μM HHMP significantly reduced LPS-induced NO production in the zebrafish larvae. Therefore, the results demonstrated that HHMP protects against LPS-induced inflammation in the zebrafish larvae model.

## 4. Materials and Methods

### 4.1. Extraction and Isolation of HHMP from J. mandshurica

*Juglans mandshurica* Maxim. fruits were purchased from the Kyung-dong herbal market in Seoul, Korea, and identified by Prof. Jong Hee Park at the College of Pharmacy, Pusan National University, Busan, Korea. The preparation, extraction, and isolation of HHMP from *J. mandshurica* fruits were described in our previous study [19]. HPLC analysis confirmed that HHMP was obtained with a purity of >95%. The chemical structure of the purified compound is shown in Figure 1A. The compound was dissolved in dimethyl sulfoxide (DMSO; Sigma-Aldrich, St. Louis, MO, USA) and used in experiments, with a final concentration of <0.01% in the culture medium.

### 4.2. Cell Culture

The murine macrophage RAW 264.7 cells were purchased from the Korean Cell Line Bank (KCLB, Korea). The cells were maintained at 37 °C in a 5% CO_2_ humidified atmosphere incubator using Dulbecco’s modified Eagle’s medium (DMEM; Welgene, Gyeongsangbuk-do, Korea), which included 10% fetal bovine serum (FBS; Welgene, Gyeongsangbuk, Korea) and 1% antibiotic-antimycotic (Gibco/BRL, Billings, MT, USA).

### 4.3. Measurement of Cell Viability

Cell viability was analyzed by altering the method descried by Cho et al. [50]. The cells were seeded in a 96-well plate at a density of 1.0 × 10^5^ cells/mL and incubated for 24 h. The cells were pre-incubated with 12.5, 25, and 50 μM of HHMP for 2 h and/or treated with 1 μg/mL lipopolysaccharide (LPS; Sigma-Aldrich, St. Louis, MO, USA) for 24 h at 37 °C. The cells were then incubated with 3-(4,5-dimethylthiazol-2-yl)-2,5-diphenyltetrazolium bromide (MTT; Amresco, Dallas, TX, USA) dissolved in phosphate-buffered saline (PBS; Gibco/BRL, Billings, MT, USA) for 2.5 h at 37 °C. After the medium was removed, the formazan was dissolved in DMSO. The absorbance was read at 540 nm using a spectrophotometer (Molecular Devices, Sunnyvale, CA, USA).

### 4.4. Nitric Oxide (NO) Assay

NO production was measured using the method described by Ham et al. [51]. The cells were incubated in a 24-well plate at a density of 1.5 × 10^5^ cells/mL for 24 h. The cells were pre-incubated with 12.5, 25, and 50 μM of HHMP for 2 h and treated with 1 μg/mL LPS for 24 h at 37 °C. A total of 100 μL of supernatant was transferred to a 96-well plate, and then 100 μL of Griess reagent (0.5% sulfanilamide, 0.05% *N*-(1-naphthyl) ethylenediamine dihydrochloride, 2.5% phosphoric acid (Sigma-Aldrich, St. Louis, MO, USA) was added, followed by incubation for 10 min in the dark. Absorbance was measured at 540 nm using a spectrophotometer (Molecular Devices, Sunnyvale, CA, USA).

### 4.5. Measurement of Prostaglandin E_2_ (PGE_2_) Production

The cells were seeded in a 24-well plate at a density of 1.5 × 10^5^ cells/mL and incubated for 24 h. The cells were pre-incubated with 12.5, 25, and 50 μM of HHMP for 2 h and treated with 1 μg/mL LPS for 24 h at 37 °C. The supernatants were collected and stored at −20 °C until analysis. The levels of PGE_2_ were quantified using a prostaglandin E_2_ parameter assay kit (R&D Systems, Minneapolis, MN, USA) according to the manufacturer’s instructions. Briefly, 150 μL of the RD5-56 calibrator diluent was added to the non-specific binding (NSB) and zero standard wells in a goat anti-mouse microplate. Next, 100 μL of the supernatant was added to the remaining wells. A total of 50 μL of primary antibody solution was added to each well, excluding the NSB wells for 2 h at room temperature, and then 50 μL of PGE_2_ conjugate was added sequentially and incubated for 16–20 h at 4 °C overnight. The plate was washed four times, and 200 μL of substrate solution was added to each well and incubated for 20 min at room temperature in the absence of light. Afterward, 50 μL of stop solution was added to each well, and the absorbance was measured at 450 nm using a spectrophotometer (Molecular Devices, San Jose, CA, USA).

### 4.6. Western Blot Analysis

The expression levels of inflammation-related protein were analyzed by altering the method described by Ham et al. [51]. The cells were seeded in 6-well plates at a concentration of 2.5 × 10^5^ cells/well and incubated for 24 h. The cells were pre-treated with various concentrations of HHMP for 2 h and 1 μg/mL LPS for 2 h at 37 °C. After the removal of media, the cells were washed twice with PBS. Radioimmunoprecipitation assay buffer (RIPA; Sigma-Aldrich, St. Louis, MO, USA) was added, and the cells were incubated for 10 min at 4 °C. The cell lysates were then collected using a scraper, vortexed six times for 1 h, and centrifuged at 15,000 rpm for 20 min at 4 °C. The protein concentration was determined using a Pierce^TM^ BCA protein assay kit (Thermo Fisher Scientific, Waltham, MA, USA). Protein samples were electrophoresed and transferred to a polyvinylidene difluoride membrane (Thermo Fisher Scientific, Waltham, MA, USA). The membranes were then blocked for 3 h at room temperature. The following primary antibodies were used at 4 °C overnight: anti-iNOS, anti-COX-2, anti-phospho-IκB-α, anti-IκB, anti-phospho-NF-κB 65, anti-phospho-NF-κB 105, anti-phospho-ERK, anti-ERK, anti-phospho-JNK, anti-JNK, anti-phospho-p38, anti-p38 (Cell Signaling Technology, USA), and anti-β-actin (Santa Cruz Biotechnology, USA). The membranes were washed with TBST (20 mM Tris, 137 mM NaCl, 0.1% Tween 20; Sigma-Aldrich, St. Louis, MO, USA) and distilled water. The following secondary antibodies were used for 2 h at room temperature: anti-rabbit IgG HRP-linked antibody and anti-mouse IgG and HRP-linked antibody (Cell Signaling Technology, Danvers, MA, USA). The bands were detected using a SuperSignal West Femto Trial kit (Thermo Fisher Scientific, Waltham, MA, USA).

### 4.7. Immunofluorescence Staining

Nuclear translocation of p65 was analyzed by altering the method described by Ham et al. [51]. The cells were seeded on a confocal slide at a concentration of 5.0 × 10^4^ cells/mL and incubated for 24 h. The cells were pre-treated with various concentrations of HHMP for 2 h at 37 °C, and 1 μg/mL LPS was added for 2 h at 37 °C. The cells were fixed with 4% paraformaldehyde (PFA; EMS, USA) for 10 min at room temperature. After the cells were washed three times with PBS, they were blocked with 1% bovine serum albumin (BSA) in PBS at room temperature for 1 h. After the cells were washed twice with PBS, they were incubated with 0.4% Triton X-100 at room temperature for 30 min. Then, they were washed thrice with PBS and incubated with p65 primary antibody (1:200; Cell Signaling Technology, Danvers, MA, USA) overnight at 4 °C. The cells were then washed thrice with PBS and incubated with Alexa Fluor488-labeled goat anti-rabbit IgG secondary antibody (1:800; Thermo Fisher Scientific, Waltham, MA, USA) for 1.5 h at room temperature in the dark. The cells were washed three times with PBS and treated with hoechst33342 (1:250; Sigma-Aldrich, St. Louis, MO, USA) for 10 min at room temperature. The cells were washed thrice with PBS and mounted using Fluoromount-G (SouthernBiotech, Birmingham, AL, USA). Fluorescence was analyzed by using an LSM 700 Zeiss confocal laser scanning microscope (Carl Zeiss, Jena, Germany).

### 4.8. Maintenance of Adult Zebrafish and Survival Rate Measurement

Adult zebrafish were maintained under the following conditions: the zebrafish were fed twice per day for 6 days a week. The day before the experiment began, zebrafish were randomly selected for mating at a female-to-male ratio of 1:2. Embryos were obtained by mating and spawning. The embryos were obtained from the breeding cages and moved to a Petri dish containing 1 mg/mL methylene blue solution. After disinfection for 1.2 h, the methylene blue solution was changed to fresh embryo media (600 mg/l red sea salt in distilled water). The survival rate was measured for 7 dpf (days post fertilization) in the presence of various concentrations of HHMP (6.25, 12.5, 25, and 50 μM). The experiments were approved by the Animal Care and Use Committee of the Korea Basic Science Institute, Korea (KBSI-20-29).

### 4.9. Maintenance of NO Production in Zebrafish Larvae

NO production in zebrafish larvae was measured by modifying the method described by Ko et al. [52]. After 7–9 hpf (hours post fertilization), the zebrafish embryos were pre-treated with 50 μM HHMP and then treated with 10 μg/mL LPS. Zebrafish larvae at 72 hpf were transferred to a 24-well plate and treated with the specific fluorescence dye diaminofluorescein-FM diacetate (DAF-FM DA; 10 μM; Sigma-Aldrich, St. Louis, MO, USA) for 3 h at 37 °C to measure NO production. The zebrafish larvae were washed three times and photographed using an LSM 700 Zeiss confocal laser scanning microscope after anesthetization with 0.03% ethyl 3-aminobenzoate methanesulfonate (MS-222; Sigma-Aldrich, St. Louis, MO, USA). The fluorescence intensity was quantified using ImageJ software (National Institute of Health, Bethesda, MD, USA).

## 5. Statistical Analysis

The data were analyzed by one-way ANOVA with Tukey’s post hoc test. Statistical significance was set at *p* < 0.05. All statistical tests were performed using GraphPad PRISM software (version 8.0; GraphPad, San Diego, CA, USA). The data are expressed as the mean ± standard deviation (SD).

## 6. Conclusions

In conclusion, our present study revealed that HHMP inhibited LPS-induced inflammatory mediators such as NO and PGE_2_ in RAW 264.7 cells by blocking the phosphorylation of NF-κB and MAPK signaling pathways. Moreover, HHMP suppressed LPS-induced NO generation in the zebrafish larvae model. Thus, HHMP may be a potential therapeutic agent that can be used for the development of functional foods or medicinal products to treat various inflammation-related disorders.

## Figures and Tables

**Figure 1 pharmaceuticals-14-00771-f001:**
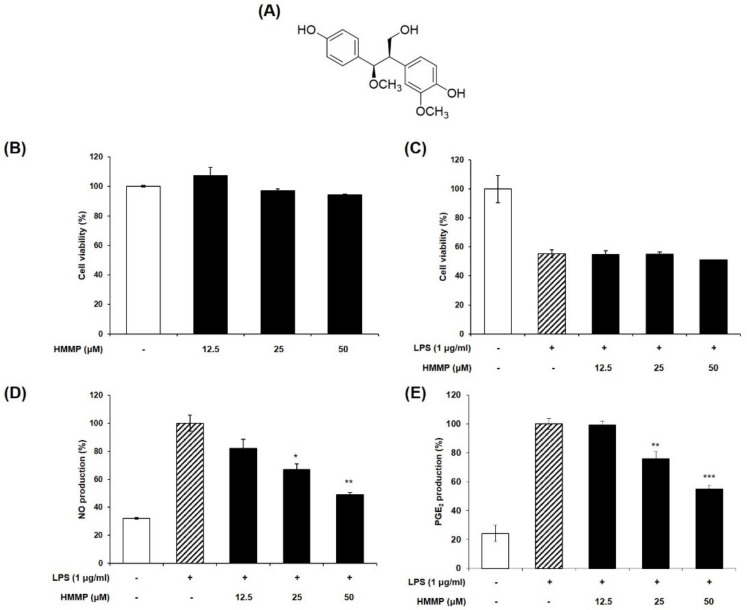
(**A**) Chemical structure of HHMP, (**B**,**C**) cell viability in presence or absence of LPS, (**D**,**E**) effect of HHMP on LPS-induced NO and PGE_2_ production in RAW 264.7 cells. Cells were pre-treated with HHMP (12.5, 25, and 50 μM) and then treated with LPS for 24 h. NO production was measured using Griess reagent and PGE_2_ production was analyzed using an ELISA kit. Cell viability was assessed with an MTT assay. +: added in the cells; -: not added in the cells. All experiments were performed in triplicate, and the results are expressed as the mean ± standard deviation (SD); * *p* < 0.05, ** *p* < 0.01, *** *p* < 0.001, and **** *p* < 0.0001 compared with the LPS-stimulated group.

**Figure 2 pharmaceuticals-14-00771-f002:**
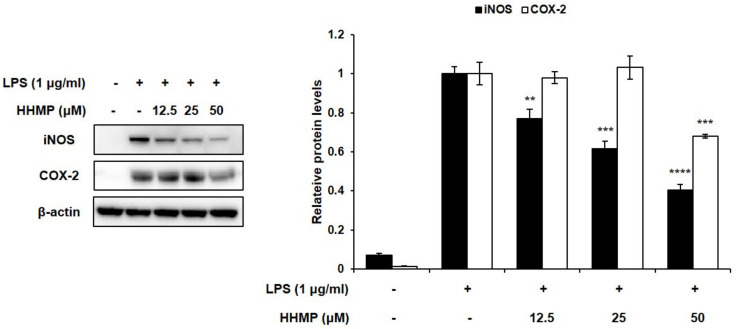
Effect of HHMP on the expression of inflammatory enzymes iNOS and COX-2 in LPS-stimulated RAW 264.7 cells. Cells were pre-treated with HHMP (12.5, 25, and 50 μM) and then treated with LPS for 24 h. Protein levels were determined using Western blot analysis. +: added in the cells; -: not added in the cells. All experiments were performed in triplicate, and the results are expressed as the mean ± standard deviation (SD); * *p* < 0.05, ** *p* < 0.01, *** *p* < 0.001, and **** *p* < 0.0001 compared with the LPS-stimulated group.

**Figure 3 pharmaceuticals-14-00771-f003:**
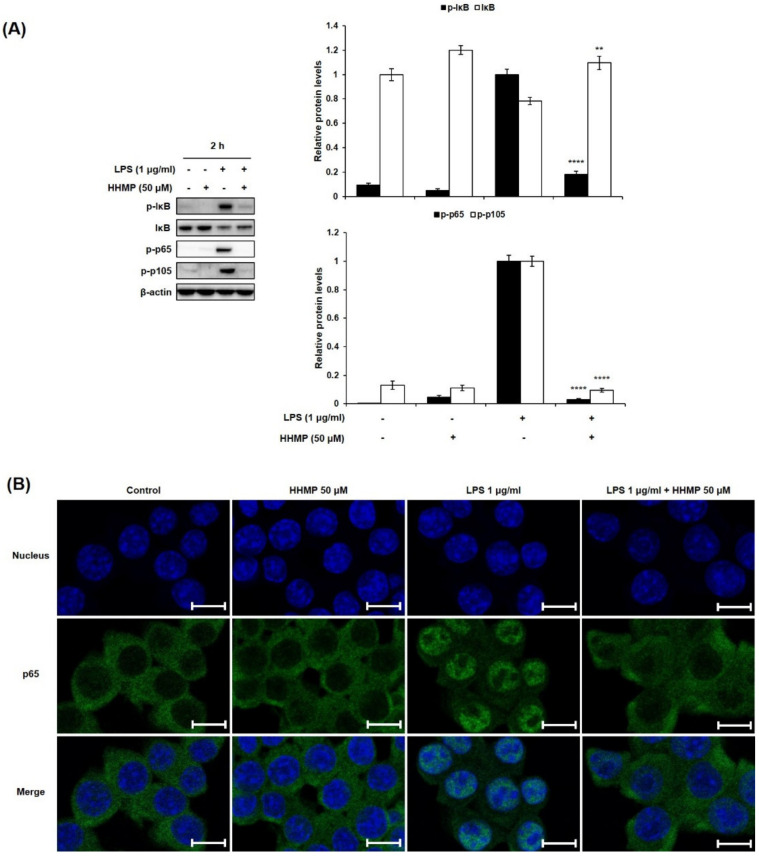
Effect of HHMP on LPS-induced (**A**) NF-κB activation and (**B**) nuclear translocation of p65 in RAW 264.7 cells. Cells were pre-treated with HHMP (12.5, 25, and 50 μM) and then treated with LPS for 2 h. Protein levels were determined using Western blot analysis. Nuclear translocation was analyzed using immunofluorescence (IF) staining and photographed using an LSM 700 Zeiss confocal laser scanning microscope. +: added in the cells; -: not added in the cells. Scale bar: 10 μm. All experiments were performed in triplicate, and the results are expressed as the mean ± standard deviation (SD); * *p* < 0.05, ** *p* < 0.01, *** *p* < 0.001, and **** *p* < 0.0001 compared with the LPS-stimulated group.

**Figure 4 pharmaceuticals-14-00771-f004:**
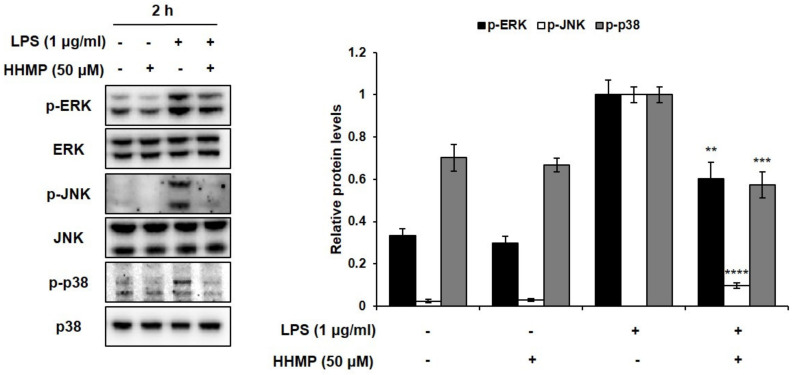
Effect of HHMP on LPS-induced MAPK activation in RAW 264.7 cells. Cells were pre-treated with HHMP (12.5, 25, and 50 μM) and then treated with LPS for 2 h. Protein levels were determined using Western blot analysis. +: added in the cells; -: not added in the cells. All experiments were performed in triplicate, and the results are expressed as the mean ± standard deviation (SD); * *p* < 0.05, ** *p* < 0.01, *** *p* < 0.001, and **** *p* < 0.0001 compared with the LPS-stimulated group.

**Figure 5 pharmaceuticals-14-00771-f005:**
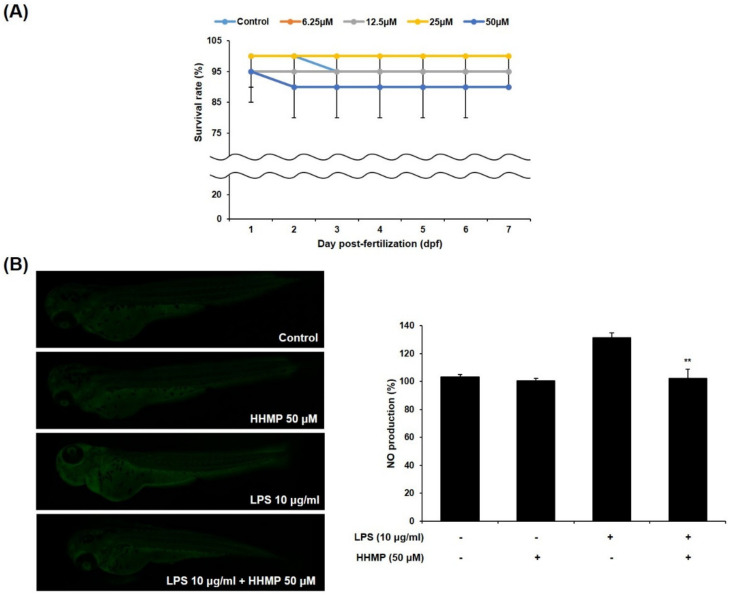
Effect of HHMP on (**A**) survival rate and (**B**) NO production in zebrafish larvae. Survival rate of zebrafish larvae was measured for 7 dpf after treatment with HHMP (6.25, 12.5, 25, and 50 μM). NO production of zebrafish larvae was assessed using an LSM 700 Zeiss confocal laser scanning microscope after treatment with 50 μM of HHMP in presence or absence of LPS for 72 hpf. Fluorescence intensity was quantified using ImageJ software. +: added in the cells; -: not added in the cells. All experiments were performed in triplicate, and the results are expressed as the mean ± standard deviation (SD); * *p* < 0.05, ** *p* < 0.01, *** *p* < 0.001, and **** *p* < 0.0001 compared with the LPS-stimulated group.

## Data Availability

Data is contained within the article.

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
