# Peer review of "Anti-Inflammatory Activity of 4-((1R,2R)-3-Hydroxy-1-(4-hydroxyphenyl)-1-methoxypropan-2-yl)-2-methoxyphenol Isolated from Juglans mandshurica Maxim. in LPS-Stimulated RAW 264.7 Macrophages and Zebrafish Larvae Model"

_pharmaceuticals, 2021, doi:10.3390/ph14080771_

Round 1
Reviewer 1 Report
In this article the authors evaluate the potential of HHMP as a natural anti-inflammatory agent. The studies performed showed that the compound inhibited LPS-induced inflammatory mediators such as NO and PGE2 by blocking the phosphorylation of NF-κB and MAPK pathways, and that it suppressed LPS-induced NO generation in the zebrafish larvae.
I have some comments about the paper:
-Since the compound used in this study, 2-(4-hydroxy-3-methoxyphenyl)-1-(4-hydroxyphenyl)-1-methoxy-3-propanol (HHMP), has two stereogenic centres there is the possibility of diastereomers. Since HHMP has been isolated from a plant it is probable that the compound has a defined stereochemistry, as it is frequent in natural products, but the stereochemistry of the compound must be mentioned in the manuscript. Did the authors test a mixture of isomers? If so, it should be interesting to separate the isomers and evaluate them separately to see the difference between the racemic mixture and each of the enantiomers.
-Please note that reference 31 couldn’t be found in the text.
-And reference 26 is cited after references 27, 28, 29 and 30 (it doesn’t appear earlier so it should be numbered accordingly).
Author Response
In this article the authors evaluate the potential of HHMP as a natural anti-inflammatory agent. The studies performed showed that the compound inhibited LPS-induced inflammatory mediators such as NO and PGE2 by blocking the phosphorylation of NF-κB and MAPK pathways, and that it suppressed LPS-induced NO generation in the zebrafish larvae.
I have some comments about the paper:
-Since the compound used in this study, 2-(4-hydroxy-3-methoxyphenyl)-1-(4-hydroxyphenyl)-1-methoxy-3-propanol (HHMP), has two stereogenic centres there is the possibility of diastereomers. Since HHMP has been isolated from a plant it is probable that the compound has a defined stereochemistry, as it is frequent in natural products, but the stereochemistry of the compound must be mentioned in the manuscript. Did the authors test a mixture of isomers? If so, it should be interesting to separate the isomers and evaluate them separately to see the difference between the racemic mixture and each of the enantiomers.
: Thank you for your comment. We have reported a defined stereochemistry in our previous work. HHMP is a novel compound isolated from Juglans mandshurica in threo isomer. Please refer reference 19 in the manuscript (Phytochemistry 137 (2017) 87-93) of compound 5. We also added the reference in the context.
-Please note that reference 31 couldn’t be found in the text.
: We added the reference 31 in Line 196 (Page 7, Discussion part).
-And reference 26 is cited after references 27, 28, 29 and 30 (it doesn’t appear earlier so it should be numbered accordingly).
: We revised the reference number line 180~186 (page 6, Discussion part) and line 396~405 (page 11, References part).
*This document certifies that the attached paper below has been edited to ensure that the language is clear and free of errors.

Reviewer 2 Report
The manuscript by Chao and colleagues reports on the anti-inflammatory properties of a new phenylpropanoid compound, 2-(4-hy-21 droxy-3-methoxyphenyl)-1-(4-hydroxyphenyl)-1-methoxy-3-propanol (HHMP) extracted from J. mands-22 hurica. The study was carried out in vitro in LPS-stimulated RAW 264.7 cells, and in vivo in the LPS-stimulated zebrafish larvae.
In summary, the study demonstrates that the mechanism of the HHMP anti-inflammatory activity is mediated by the inhibition of the MAPK/NF‐κB signaling pathway. This inhibition considerably reduces the LPS-induced NO and PGE2 generation 77 in RAW 264.7 cells, and the LPS-induced NO production in zebrafish larvae. Based on these findings, the authors indicate the HHMP as a potential therapeutic to treat inflammation-related disease.
The experiments were well designed and performed, and the conclusions are overall fairly supported by the data obtained. The manuscript is adequately presented and written. Just a few comments that authors could consider to improve the quality of their studies.
- Authors should better introduce and describe the MAPK/NF‐κB signaling pathway, explaining more in detail the role of each component of the complex (p105, p65) specifically in the development of anti-inflammatory response (considering that MAPK/NF‐κB are involved in multiple pathways). Also, explaining why some components are studied considering the nuclear translocation could improve the comprehension to the reader. For each investigation undertaken, authors indicate the candidate target under examination but the reason why those targets are acknowledged for a given conclusion is not well explained. I suggest spending some words just before presenting assays and results obtained.
- The authors do not consider any aspect concerning the upstream mechanism by which the LPS-induced MAPK/NF‐κB signaling pathway is repressed by the HHMP. Does HHMP interact with some membrane receptors that activate an anti-inflammatory response? Is HHMP up taken by cells? The manuscript would be improved considering this point, at least discussing it in terms of hypothetical mechanism.
- Here are some minor observations, line 95: “We demonstrated whether the inhibitory….” Whether is not followed by … line 174: “We had isolated” sounds a “narrative tense”. I find it better to use We isolated or have isolated.
Author Response
The manuscript by Chao and colleagues reports on the anti-inflammatory properties of a new phenylpropanoid compound, 2-(4-hy-21 droxy-3-methoxyphenyl)-1-(4-hydroxyphenyl)-1-methoxy-3-propanol (HHMP) extracted from J. mands-22 hurica. The study was carried out in vitro in LPS-stimulated RAW 264.7 cells, and in vivo in the LPS-stimulated zebrafish larvae.
In summary, the study demonstrates that the mechanism of the HHMP anti-inflammatory activity is mediated by the inhibition of the MAPK/NF‐κB signaling pathway. This inhibition considerably reduces the LPS-induced NO and PGE2 generation 77 in RAW 264.7 cells, and the LPS-induced NO production in zebrafish larvae. Based on these findings, the authors indicate the HHMP as a potential therapeutic to treat inflammation-related disease.
The experiments were well designed and performed, and the conclusions are overall fairly supported by the data obtained. The manuscript is adequately presented and written. Just a few comments that authors could consider to improve the quality of their studies.
Authors should better introduce and describe the MAPK/NF‐κB signaling pathway, explaining more in detail the role of each component of the complex (p105, p65) specifically in the development of anti-inflammatory response (considering that MAPK/NF‐κB are involved in multiple pathways). Also, explaining why some components are studied considering the nuclear translocation could improve the comprehension to the reader. For each investigation undertaken, authors indicate the candidate target under examination but the reason why those targets are acknowledged for a given conclusion is not well explained. I suggest spending some words just before presenting assays and results obtained.
: Thank you for your valuable comment. We added the detailed description of MAPK/NF‐κB signaling pathway as you suggested.
The authors do not consider any aspect concerning the upstream mechanism by which the LPS-induced MAPK/NF‐κB signaling pathway is repressed by the HHMP. Does HHMP interact with some membrane receptors that activate an anti-inflammatory response? Is HHMP up taken by cells? The manuscript would be improved considering this point, at least discussing it in terms of hypothetical mechanism.
: Upstream signaling pathway is well known through previous studies. We attempted to isolate HHMP but did not succeed in isolating sufficient material to perform the experiment. However, our study demonstrated that HHMP directly inhibits the LPS-induced activation of NF‐κB and MAPK. We hope you understand.
Here are some minor observations, line 95: “We demonstrated whether the inhibitory….” Whether is not followed by … line 174: “We had isolated” sounds a “narrative tense”. I find it better to use We isolated or have isolated.
: We revised the sentences as you suggested.
*This document certifies that the attached paper below has been edited to ensure that the language is clear and free of errors.

Reviewer 3 Report
Cho et al. previously isolated and elucidated a phenolic compound, namely 2-(4-hydroxy-3-methoxyphenyl)-1-(4-hydroxyphenyl)-1-methoxy-3-propanol, within a phytochemical study of the ingredients of the fruits of Juglans mandshurica which is indigenous to Korea and China. By applying various in vitro assays and an in vivo test model the authors prove anti-inflammatory effects of this phenol in the present study.
The whole manuscript is written in an excellent and fluent style. There are no spelling mistakes (except one in the heading). This contribution is of high value for the international readership and may lead to new therapeutic agents in the future for the treatment of inflammations. The reference list consists of 44 entries.
There are only some minor issues which need to be addressed:
Line 1 (the heading): spelling mistake, compound name should be 2-(4-hydroxy-3-methoxyphenyl)-1-(4-hydroxyphenyl)-1-methoxy-3-propanol, i.e. elimination of „a“
Line 87-93: The explanation for Figure 1 is partly not correct. It should be „(A) Chemical structure of HHMP, (D,E) effect of HHMP on LPS-induced NO and PGE2 production, and (B,C) cell variabilty in presence or absence of LPS in RAW 264.7 cells.“ Please check this again!
Line 101: The authors mention, that HHMP reduces NO and PGE2 production by inhibiting LPS-induced expression of the inflammatory enzymes iNOS and COX-2 in RAW 264.7 cells. Regarding Figure 2 this is obviously true for iNOS, but for COX-2, the results do not demonstrate an inhibition at the concentration of 25 µM. Please discuss this finding and rephrase the sentence on the results for COX-2!
Line 161/Figure 4: The scale for (A) should be adapted, i.e, the region e.g. from 75 % to 105 % should be enlarged. The graph in its present form is difficult to understand and to follow.
Line 330: I think that an antiinflammatory agent should lead to the development of a medicinal product used in therapy. Therefore I propose either to delete „functional foods“ or at least to add „or medicinal products“.
In conclusion the studies were well perfomed and the manuscript is very well prepared and is of great interest for the international audience. Only a few minor corrections and additions are needed which are easily performed.
Author Response
Cho et al. previously isolated and elucidated a phenolic compound, namely 2-(4-hydroxy-3-methoxyphenyl)-1-(4-hydroxyphenyl)-1-methoxy-3-propanol, within a phytochemical study of the ingredients of the fruits of Juglans mandshurica which is indigenous to Korea and China. By applying various in vitro assays and an in vivo test model the authors prove anti-inflammatory effects of this phenol in the present study.
The whole manuscript is written in an excellent and fluent style. There are no spelling mistakes (except one in the heading). This contribution is of high value for the international readership and may lead to new therapeutic agents in the future for the treatment of inflammations. The reference list consists of 44 entries.
There are only some minor issues which need to be addressed:
Line 1 (the heading): spelling mistake, compound name should be 2-(4-hydroxy-3-methoxyphenyl)-1-(4-hydroxyphenyl)-1-methoxy-3-propanol, i.e. elimination of „a“
: Thank you for your comment. We revised the compound name as you suggested.
Line 87-93: The explanation for Figure 1 is partly not correct. It should be „(A) Chemical structure of HHMP, (D,E) effect of HHMP on LPS-induced NO and PGE2 production, and (B,C) cell variabilty in presence or absence of LPS in RAW 264.7 cells.“ Please check this again!
: We revised the Figure 1 legend as you suggested (line 87, 88).
Line 101: The authors mention, that HHMP reduces NO and PGE2 production by inhibiting LPS-induced expression of the inflammatory enzymes iNOS and COX-2 in RAW 264.7 cells. Regarding Figure 2 this is obviously true for iNOS, but for COX-2, the results do not demonstrate an inhibition at the concentration of 25 µM. Please discuss this finding and rephrase the sentence on the results for COX-2!
: Thank you for your valuable comment. We conducted the experiment three times to demonstrate the effect of HHMP on expression of COX-2. We proved that 12.5 and 25 μM of HHMP showed no significant difference compared with LPS-treated group. We don’t know why the tendencies of PGE2 production and COX-2 expression are different. However, 50 μM of HHMP significantly inhibited the LPS-induced expression of COX-2. Previous studies have shown that low concentrations result in different tendencies of PGE2 production and COX-2 expression [1-3]. Therefore, our results showed that low concentration (12.5 and 25 μM) of HHMP inhibits PGE2 production without affecting COX-2 protein.
[1] Hong, J.M.; Kwon, O.K.; Shin, I.S.; Jeon, C.M.; Shin, N.R.; Lee, J.; Park, S.H.; Bach, T.T.; Hai, D.V.; Oh, S.E.; Han, S.B.; Ahn, K.S., Anti-inflammatory effects of methanol extract of Canarium lyi C.D. Dai & Yakovlev in RAW 264.7 macrophages and a murine model of lipopolysaccharide-induced lung injury. Int. J. Mol. Med. 2015, 35, 1403-1410.
[2] Kim, H.S.; Park, J.W.; Kwon, O.K.; Kim, J.H.; Oh, S.R.; Lee, H.K.; Bach, T.T.; Quang, B.H.; Ahn, K.S., Anti-inflammatory activity of a methanol extract from Ardisia tinctoria on mouse macrophages and paw edema. Mol. Med. Rep. 2014, 9, 1388-1394.
[3] Kim, J.; Kim, H.; Choi, H.; Jo, A.; Kang, H.; Yun, H.; Im, S.; Choi, C., Anti-inflammatory effects of a Stauntonia hexaphylla fruit extract in lipopolysaccharide-activated RAW-264.7 macrophages and rats by carrageenan-induced hind paw swelling. Nutrients 2018, 10, 110.
Line 161/Figure 4: The scale for (A) should be adapted, i.e, the region e.g. from 75 % to 105 % should be enlarged. The graph in its present form is difficult to understand and to follow.
: We revised the graph as you suggested.
Line 330: I think that an antiinflammatory agent should lead to the development of a medicinal product used in therapy. Therefore I propose either to delete „functional foods“ or at least to add „or medicinal products“.
: We added the “functional foods or medicinal products” as you suggested.
In conclusion the studies were well perfomed and the manuscript is very well prepared and is of great interest for the international audience. Only a few minor corrections and additions are needed which are easily performed.
*This document certifies that the attached paper below has been edited to ensure that the language is clear and free of errors.

Reviewer 4 Report
Overall this study was well designed and executed. The manuscript is well written, and the results are convincing. Abstract has written perfectly. However, some points need to be addressed before publications. The study's rationale: How did the author come to know that HHMP will work on inflammation. The proper rationale in the introduction section will make the manuscript attractive. Figure 1: Figure captions are not correct. Error bar in figure 1C is missing. Figure 1B is not necessary. Figure 2: Error bar is missing. The statistical level of significance is not correct. Authors need to revisit and correct it. Most importantly, why cox-2 expression was not dose-dependent? Explain it in results and discussion with proper justification and reference. Some other figures also do not have an error bar. If it is low, then it is hard to believe with such low error of any experiment. How many experiments performed exactly mention it (N= ?). Any of the methodology sections do not have a reference. If authors develop all the methodology by themselves, then give validation data in supportive information or cite references for each method. If we are following some researcher's methods even they are old, we still need to acknowledge them by citing their research articles.Author Response
Overall this study was well designed and executed. The manuscript is well written, and the results are convincing. Abstract has written perfectly. However, some points need to be addressed before publications.
The study's rationale: How did the author come to know that HHMP will work on inflammation. The proper rationale in the introduction section will make the manuscript attractive.
: Thank you for your comment. We revised the introduction section as you suggested.
Figure 1: Figure captions are not correct.
: We revised the Figure 1 legend as you suggested (line 87, 88).
(A) Chemical structure of HHMP, (B, C) cell viability in presence or absence of LPS, (D, E) effect of HHMP on LPS-induced NO and PGE2 production in RAW 264.7 cells. Error bar in figure 1C is missing.
Figure 1B is not necessary.
: We conducted cell viability of HHMP and demonstrated that it did not affect the cell viability. Therefore, we added Figure 1(B) graph.
Figure 2: Error bar is missing. The statistical level of significance is not correct. Authors need to revisit and correct it. Most importantly, why cox-2 expression was not dose-dependent? Explain it in results and discussion with proper justification and reference. Some other figures also do not have an error bar. If it is low, then it is hard to believe with such low error of any experiment.
: Thank you for pointing out our mistakes. We recalculated because there was the mistake in the calculating process. We performed the experiment three times to prove the effect of HHMP on LPS-induced COX-2 expression. As a result, 12.5 and 25 μM of HHMP showed no significant difference compared with LPS-treated group. And, we revised other figures after recalculating again. We really appreciate your help.
How many experiments performed exactly mention it (N= ?).
: We mentioned “three individual experiments (n=3)” in figure legends as you suggested.
Any of the methodology sections do not have a reference. If authors develop all the methodology by themselves, then give validation data in supportive information or cite references for each method. If we are following some researcher's methods even they are old, we still need to acknowledge them by citing their research articles.
: We mentioned the references in the Materials and methods part as you suggested (Page 8, line 245, 254, 278, 299, 327).
*This document certifies that the attached paper below has been edited to ensure that the language is clear and free of errors.
